# Quantifying the gender gap in the HIV care cascade in southern Mozambique: We are missing the men

Elisa Lopez-Varela[1,2]☉*, Orvalho Augusto[1,3]☉, Laura Fuente-Soro[1,2], Charfudin Sacoor[1], Ariel Nhacolo[1], Isabelle Casavant[4], Esmeralda Karajeanes[5], Paula Vaz[5], Denise Naniche[1,2]

1 Centro de Investigação em Saúde de Manhiça (CISM), Maputo, Mozambique, 2 ISGlobal, Instituto de Salud Global de Barcelona, Hospital Clínic—Universitat de Barcelona, Barcelona, Spain, 3 Facultade de Medicina, Universidade Eduardo Mondlane, Maputo, Mozambique, 4 Centers for Disease Control (CDC), Maputo, Mozambique, 5 Fundação Ariel Glaser Contra o SIDA Pediátrico, Maputo, Mozambique

☉ These authors contributed equally to this work.
* elisa.lopez@isglobal.org

**Data Availability Statement:** The minimum dataset underlying the results of this study are available upon request due to ethical restrictions imposed by the Ministry of Health of Mozambique

## Abstract

### Background

HIV-infected men have higher rates of delayed diagnosis, reduced antiretroviral treatment (ART) retention and mortality than women. We aimed to assess, by gender, the first two UNAIDS 90 targets in rural southern Mozambique.

### Methods

This analysis was embedded in a larger prospective cohort enrolling individuals with new HIV diagnosis between May 2014-June 2015 from clinic and home-based testing (HBT). We assessed gender differences between steps of the HIV-cascade. Adjusted HIV-community prevalence was estimated using multiple imputation (MI).

### Results

Among 11,773 adults randomized in HBT (7084 female and 4689 male), the response rate before HIV testing was 48.7% among eligible men and 62.0% among women (p<0.001). MI did not significantly modify all-age HIV-prevalence for men but did decrease prevalence estimates in women from 36.4% to 33.0%. Estimated proportion of HIV-infected individuals aware of their status was 75.9% for men and 88.9% for women. In individuals <25 years, we observed up to 22.2% disparity in awareness of serostatus between genders. Among individuals eligible for ART, similar proportions of men and women initiated treatment (81.2% and 85.9%, respectively). Fourfold more men than women were in WHO stage III/IV AIDS at first clinical visit. Once on ART, men had a twofold higher 18-month loss to follow-up rate than women.

### Conclusion

The contribution of missing HIV-serostatus data differentially impacted indicators of HIV prevalence and of achievement of UNAIDS targets by age and gender and men were

ethics review board. Public sharing of the data would breach the ethics review board's data transfer agreement (DTA). Additionally, the data contains personal identifiers and sensitive information regarding HIV infection and treatment, as well as other infections. Interested researchers may contact LLorenc Quinto llorenc.quinto@isglobal.org with their proposed analysis.

**Funding:** Supported by the President's Emergency Plan for AIDS Relief (PEPFAR) through the Centers for Disease Control and Prevention (CDC) under the terms of CoAg GH000479 (Scaling-up HIV counseling and testing services in a rural population by strengthening the health demographic surveillance system, in Manhiça, Mozambique). The findings and conclusions in this report are those of the author(s) and do not necessarily represent the official position of the CDC.

**Competing interests:** The authors have declared that no competing interests exist.

missing long before the second 90. Increased efforts to characterize missing men and their needs will and their needs will allow us to urgently address the barriers to men accessing care and ensure men are not left behind in the UNAIDS 90-90-90 targets achievement.

## Introduction

In 2014, UNAIDS set the ambitious global strategy of reaching the 90-90-90 targets to end the HIV epidemic by 2020: 90% of people living with HIV (PLWHIV) will know their HIV status, 90% of those will be on antiretroviral therapy (ART), and of those, 90% will reach viral suppression. Nearly 70% of the 36.7 million PLWHIV live in sub-Saharan Africa, where most countries have adopted the WHO universal treatment approach [1]. However, limited resources, sociocultural norms, and high HIV prevalence in certain sub-Saharan African regions present challenges to efficient and equitable HIV diagnosis and treatment.

According to the most recent UNAIDS global HIV data report on the progress towards these targets, men are lagging behind at 75-74-85 compared to 84-81-87 for women [2,3]. Although the incidence of new HIV infections in sub-Saharan Africa is higher in women, HIV-positive men have a 41% higher risk of dying than HIV-positive women [4–6]. In South Africa, estimates show that 51% of women living with HIV receive ART compared to 37% of men [7]. In Mozambique, according to the latest HIV prevalence survey, 78.2% of self-reported HIV-positive women receive ART compared to 68% of men [8]. Women are more likely to have contact with the health system through maternity and child health services and thus have facilitated access to HIV diagnosis and treatment. For example, in Malawi, adult women (excluding pregnant women and those with infants) have a yearly average of 19 hours of health system interaction compared to three in men [9]. Men, however, tend to have more irregular contact with the health system and are diagnosed at later stages [10]. Reasons cited by men for lower engagement include confidentiality concerns, inconvenient hours, and perceptions that facilities provide women-centered services [11–13]. Thus, because men attend the health facility more infrequently, health system–based studies cannot accurately measure achievement of the first nor third 90 target [12]. Population-based studies are necessary to ensure that all men, women, age groups, and key populations are included. Health demographic surveillance systems (HDSS) can provide crucial metrics and identification of outcomes such as loss to follow-up (LTFU), death, and silent transfers, all of which are challenging in many regions of sub-Saharan Africa that lack updated census and tracking systems.

Population-based data are necessary to estimate the achievement of the UNAIDS targets not only to understand where attrition is occurring but also to estimate the effects of serosurvey non-response rates on prevalence and on achievement of the first 90 target [14]. In this study, we assessed the first two 90 targets in men and women in Manhiça, southern Mozambique.

## Materials and methods

### Study design and participants

The study was performed in the HDSS located in the Manhiça district, southern Mozambique, with a high HIV burden of disease [15,16]. The majority of the population are rural and most are en-gaged in subsistence farming, There is a predominance of young people and nearly half of all women and a quarter of all men are illiterate [17]. At the time of the study in 2015, this district covered a total population of approximately 174,000 [16] under demographic

surveillance with vital events updated through annual home visits [18]. The district is served by the Manhiça District Hospital (MDH) and 11 peripheral health facilities that offer free HIV services. The CD4 threshold for ART initiation at the time of the study per national guidelines was ≤350 cells per µL and ≤500 cells per µL after 2016. Patients are seen every three or six months according to ART eligibility.

This analysis was embedded in a larger prospective cohort study that consecutively enrolled patients with new HIV diagnosis between May 2014 and June 2015, tested at two clinic-based venues: voluntary counselling and testing (VCT), provider-initiated counselling and testing (PICT), and a home-based testing (HBT) campaign [18].

## Ethics

This study was approved by the Mozambican National Bioethics Committee (REF: 51/CNBS/ 13) and the Institutional Review Boards at the Centers for Disease Control and Prevention, the Barcelona Institute of Global Health, and the Centro de Investigação em Saúde de Manhiça. All participants provided written informed consent.

## Procedures

To assess the awareness of HIV positive status as a proxy for the first 90 target, we used data from the HBT campaign [18]. HBT was offered to adults randomly selected from the HDSS area 2014 enumeration served by the MDH; these adults were visited at their homes, offered HIV testing and counselling (HTC), and asked to participate in the study [18]. The HBT campaign updated those deaths and migrations not captured by the HDSS during the 12 months of HBT. HTC was conducted by finger prick rapid testing according to national guidelines. Although only the selected individual was invited to participate in the study, HTC was offered to the household. Before HBT, the randomly selected adults were crosschecked in the MDH HIV electronic patient tracking system (ePTS), a Ministry of Health managed clinical database of patients in care. Those with prior enrollment history of HIV care were not visited [18,19]. Since pregnant women followed a specific model of integrated care, they were excluded from the study cohort which assessed the HIV care cascade for non-pregnant adults. Participants eligible for HTC were those who self-reported a negative HIV test more than three months before HBT or who were unaware of their serostatus. Participants with evidence of enrollment in HIV care or who disclosed their seropositivity were encouraged to continue clinical follow-up but were not retested. A minimum of three contact attempts were made per individual before defining the status as absent, and new individuals were contacted when households could not be found, as previously described [18].

To assess the second 90 target, rates of linkage to care, ART initiation, LTFU, and mortality rates, we recruited all individuals with a new HIV diagnosis at any of the three testing modalities in the main cohort. Since this study was conducted prior to test and treat, the estimates were based on those with ART eligibility since those without ART criteria were not offered ART. The study did not influence linkage to care beyond HTC and facility-based guidance [18]. Passive follow-up measures of linkage and retention in HIV care data were obtained from clinical information registered in the ePTS, and vital status data were extracted from the HDSS.

## Outcome definitions

Responders in HBT were defined as those tested for HIV whereas no serostatus data included those who were not found at home or who chose not to participate. Awareness of HIV-positivity was defined as self-reporting HIV-positivity to the counselor or being identified through probabilistic record linkage crosschecking in the ePTS [20]; lack of previous awareness of

HIV-positivity was defined as having a new HIV diagnosis from the counselor and having no existing record in the ePTS. Unadjusted community HIV prevalence was estimated as the total number of HIV-positive individuals divided by the number of HIV-positive and HIV-negative (self-reported or tested) individuals in the HBT campaign.

Linkage to care was defined as receiving a CD4 result within three months of diagnosis. ART initiation was defined as initiating ART less than three months after reaching ART eligibility criteria. For this purpose, we considered ART eligibility and initiation based on the criteria at the first clinical visit. Patients considered LTFU on ART were those who initiated ART, and did not have a clinical visit in the previous 180 days [21].

## Statistical analysis

Descriptive characteristics were compared using $\chi^2$ or Fisher exact tests for categorical variables and Kruskal-Wallis for continuous variables. Absolute differences in the proportion of men and women (gap women to men) aware of their HIV-positivity were estimated through a GLM regression with a binomial family, identity link, and Huber-White standard errors.

HIV prevalence was calculated using two methods: unadjusted and excluding individuals with missing HIV status (complete cases), adjusted for missing HIV status data by multiple imputation (MI). We conducted MI for each gender separately. Thirty imputed datasets were generated from the MI chained equations procedures using data from the HDSS including age, education, marital status, religion, neighborhood, and wealth quintile, produced as a principal components analysis score based on the household assets. The 30 prevalence estimates were combined according to Rubin rules [22]. Fold-change in prevalence between unadjusted and adjusted values was calculated by dividing the adjusted prevalence estimate by the unadjusted prevalence estimate.

Outcomes of linkage to care and ART initiation were assessed using logistic regression and the Wald statistic and adjusted for age and HIV-testing modality. The Kaplan-Meier procedure was used to estimate the proportion of patients LTFU at 12 months on ART, and the z test was performed to assess for differences in proportions. To estimate the cumulative incidence of LTFU over the first 18 months of ART, we included patients who had initiated ART at least six months prior to the analysis (in order to meet the definition of LTFU) and applied the Fine and Gray competing risk model, with death or transfer before LTFU being the main competing events. The hazard ratio (HR) of the sub-distribution (SHR) was used as a measure of association together with the corresponding 95% CI. To account for potential confounders identified in a previous study [18], we adjusted the model for age, testing modality and WHO stage at first clinical visit. P values below 0.05 were considered statistically significant. All statistical analyses were performed using Stata 14.1 (Stata Corp, College Station, TX, USA).

## Results

### Acceptance and response rate in the HBT campaign

The study population (11,773 adults) was predominantly women (60.2%) and had a median age of 34.2 years (35.6 years for females and 32.3 for males). A total of 850 adults were identified as HIV-positive through crosschecking with ePTS and were ineligible, and 42 were excluded owing to missing household location and/or geocoordinate data (Fig 1). Field counselors approached 10,881 individuals for HBT in the Manhiça HDSS: 69.0% (95% CI 67.6 to 70.4) of men and 79.6% (95% CI 78.6 to 80.6) of women were found at home (p<0.001, Fig 2). The proportion of people not found due to death or other reasons was similar between men and women [18]. Absences from the household after three visits were significantly higher in men than women (12.6% vs. 4.5%, respectively; p<0.0001) as were absences due to migration

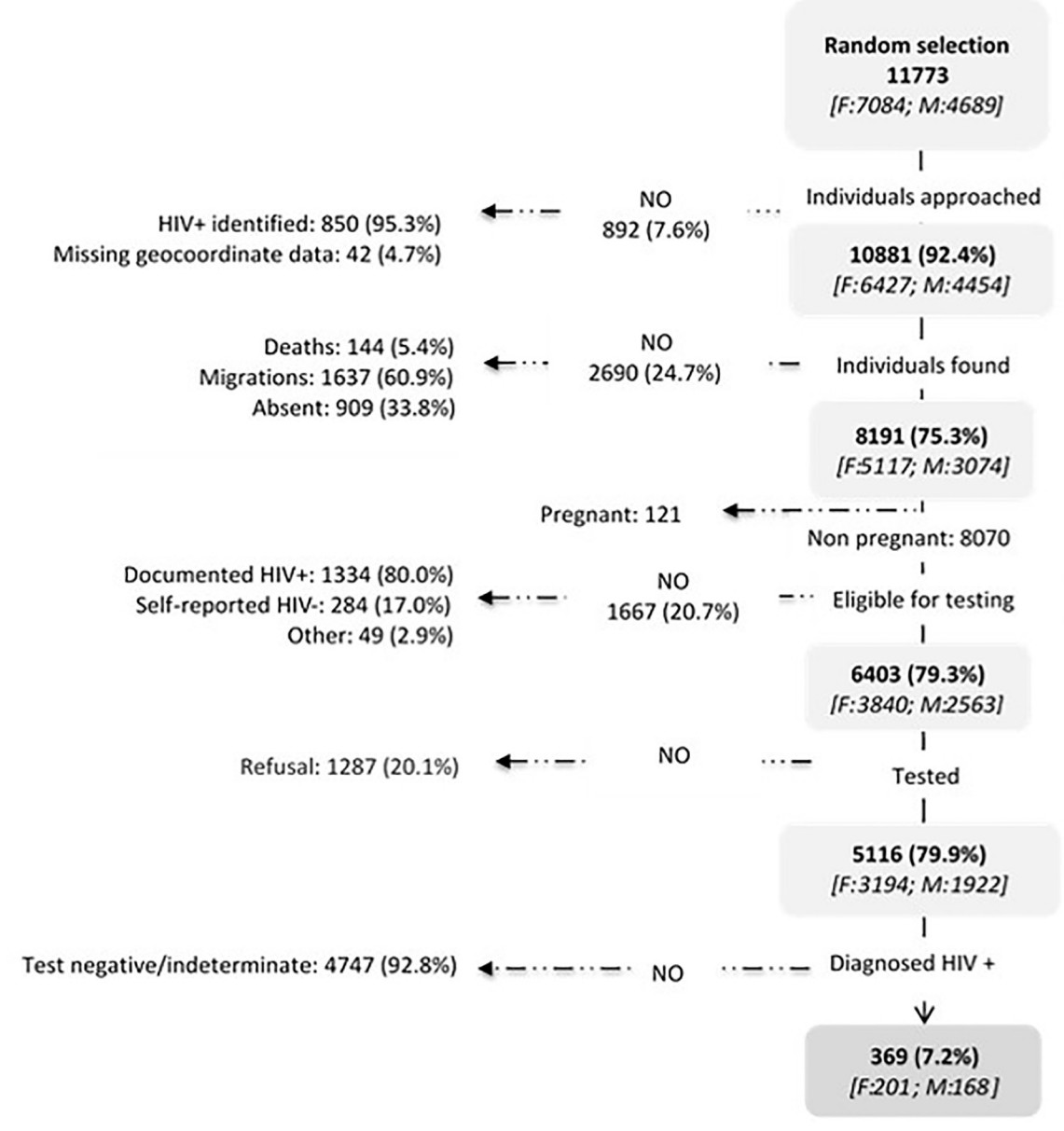

**Fig 1. Study profile for HIV home-based testing in rural southern Mozambique.** Percentage are calculated over the previous step. The shaded boxes refer to those participants included in each step and the arrows to the left give the reasons for non-inclusion. The number of female and male participants at each step are shown in brackets.

(16.4% vs. 14.1%, respectively; p = 0.01). Of those eligible for HIV testing, 75.0% (95% CI 73.3 to 76.7) of men and 83.2% (95% CI 82.0 to 84.3) of women accepted HIV testing (Fig 2). Thus, the overall response rate before HIV testing, after accounting for absences and non-participation, was 48.7% (1922/3943) among eligible men and 62.0% (3194/5150) among eligible women (p<0.0001; Table 1).

## Adjustment of community HIV prevalence estimates

Responders differed significantly in gender and age from individuals without HIV status data due to absenteeism, migration and non-participation. (Table 1). Of the 5,116 individuals who

received HIV testing (Fig 1), the crude prevalence estimate of new HIV diagnoses was 7.2% (95% CI 6.5 to 7.9) and was higher in men (168/1922 [8.7%]; 95% CI 7.5 to 10.0) than in women (201/3194 [6.3%]; 95% CI 5.5 to 7.2; p<0.001). The unadjusted HIV community prevalence, among those with recorded HIV status, was estimated to be 33.6% (95% CI 32.5 to 34.6) and varied significantly by gender and age (Fig 3). MI did not significantly modify the all-ages HIV prevalence for men although it did decrease prevalence estimates among women from 36.4 (95% CI 35.1 to 37.7) to 33.0% (95% CI 31.6 to 34.3; Fig 3). The effect of MI on HIV prevalence varied across ages. Among the younger population (age<25), in both women and men, MI increased HIV prevalence estimates by two times in men and by 1.4 in women. On the other hand, MI significantly decreased the prevalence estimate in men aged 25–54 and women >35 years. The greatest decrease in prevalence estimate was observed in men aged 35–44 and women aged 45–54 years: MI-adjusted HIV prevalence was 0.3 and 0.2 times lower, respectively.

## Knowledge of HIV positive serostatus in men and women

Of the 5,116 individuals who received HIV testing (Fig 1), the crude prevalence estimate of new HIV diagnoses was 7.2% (95% CI 6.5 to 7.9) and was higher in men (168/1922 [8.7%]; 95% CI 7.5

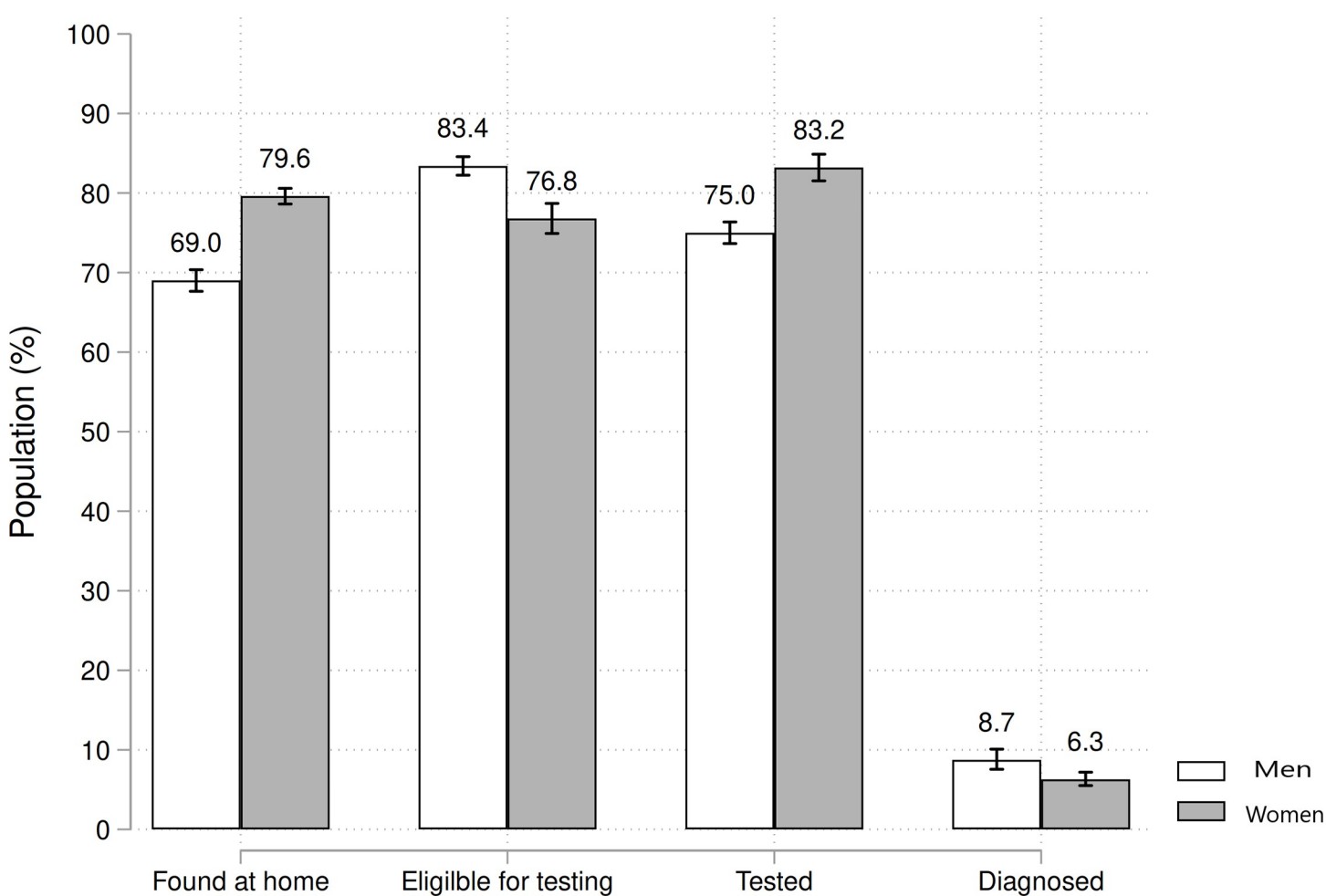

**Fig 2. Individuals reached for HIV testing during a home-based testing campaign in rural southern Mozambique.** Proportions were calculated as the number of people achieving each step divided by the numerator of the previous step.

**Table 1. Age distribution of the population found and tested for HIV (responders) compared to those not found and/or not tested (non-responders) by gender in rural southern Mozambique.**

| Characteristic | Females | | | Males | | |
|---|---|---|---|---|---|---|
| | Responders N (%) | Non Responders N (%) | Response rate (%) | Responders N (%) | Non Responders N (%) | Response rate (%) |
| All | 3194(100.0) | 1956(100.0) | 62.0 | 1922(100.0) | 2021(100.0) | 48.7 |
| Age category | | | | | | |
| <25 | 691(21.6) | 583(29.8) | 45.8 | 648(33.7) | 589(29.1) | 47.6 |
| 25–34 | 657(20.6) | 553(28.3) | 45.7 | 444(23.1) | 693(34.3) | 60.9 |
| 35–44 | 488(15.3) | 322(16.5) | 39.8 | 254(13.2) | 354(17.5) | 58.2 |
| 45–54 | 380(11.9) | 160(8.2) | 29.6 | 153(8.0) | 186(9.2) | 54.9 |
| 55–64 | 361(11.3) | 117(6.0) | 24.5 | 171(8.9) | 109(5.4) | 38.9 |
| >64 | 616(19.3) | 219(11.2) | 26.2 | 251(13.1) | 88(4.4) | 26.0 |
| Median (IQR) | | | | | | |

[a]Total N = 9093, which includes all 10881 individuals approached minus those pregnant (N = 121) and not eligible (N = 1667). Non responders (N = 3977) include those participants not found (N = 1287) plus those not tested (N = 1287), while responders (N = 5116) were those who were tested.

[b]Denominator used for age, 9087.

to 10.0) than in women (201/3194 [6.3%]; 95% CI 5.5 to 7.2; p<0.001) (Fig 2). The unadjusted HIV community prevalence, among those with recorded HIV status, was estimated to be 33.6% (95% CI 32.5 to 34.6) and varied significantly by gender and age (Fig 3).To account for individuals with missing HIV status, after adjustment by MI, we estimated that 75.9% (95% CI 72.4 to 79.5) of men and 88.9% (95% CI 87.5 to 90.4) of women were aware of their status (p<0.001; Fig 4A). We observed a total 13.0% disparity (95% CI 9.2 to 16.8) in awareness of serostatus between genders (Fig 4 and S1 Fig). Men younger than 35 years were significantly less aware of their HIV positive status as compared to same-age women (p<0.001) with a 25.7% (95% CI 13.7 to 39.2) gap in awareness for men under 25 years old as compared to women (Fig 4). In patients over 35 years-old, similar proportions of men and women were aware of their HIV-positive status.

## Linkage to care among men and women with a new HIV diagnosis

Thirty-eight percent of our PLHIV cohort were diagnosed by PICT and 29% by VCT,. Overall, men were as likely as women to be linked to care within three months' post-diagnosis (42.4% vs. 44.7%; logistic regression p = 0.52, adjusted for testing modality and age). However, at first clinical visit, fourfold more men were in WHO stage III/IV AIDS than women (9.8% vs 2.4%; p<0.0001), and 20.6% of men and 11.4% of women had advanced HIV disease with CD4 <100 cells per μL (p = 0.001). Of those linked to care, men had a median 278 CD4 cells per μL (interquartile range [IQR], 138–424), and women had a median 305 cells per μL (IQR, 179–466; p = 0.03). Among individuals meeting the criteria for ART initiation, similar proportions of men and women were on ART at three months post-diagnosis (81.2% 125/154 vs. 85.9% 158/184, respectively; logistic regression Wald test p = 0.20, adjusted for testing modality and age). Similarly, after stratification by testing modality, there was no difference between proportions of men and women on ART (p = 0.14, p = 0.79 and p = 0.95 for VCT, PICT and HBT respectively). The proportion on ART among all diagnosed individuals irrespective of ART eligibility was 25.2% and did not differ between men and women (p = 0.95).

## Retention in care among men and women with a new HIV diagnosis

At 12 months post-ART initiation, twofold more men than women were LTFU (22.2% [95% CI 16.8 to 28.9] vs 10.9% [95% CI 7.6 to 15.5], respectively; p = 0.01). On univariate analysis,

**A**

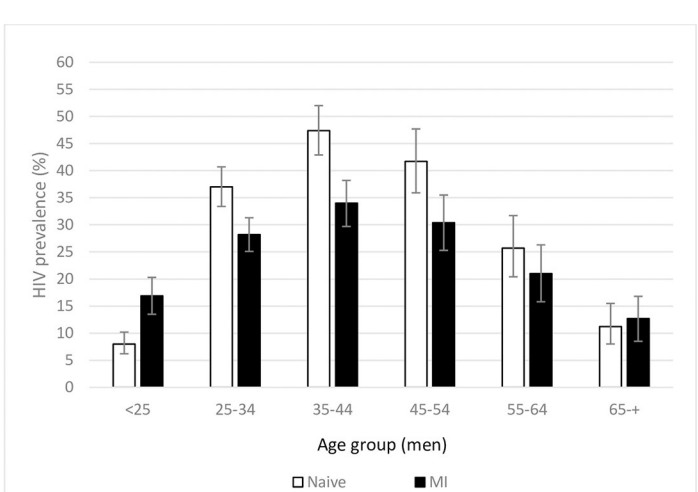

**B**

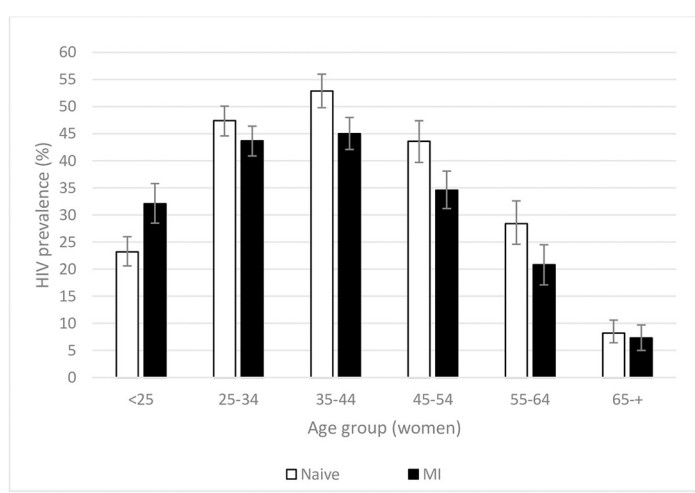

**C**

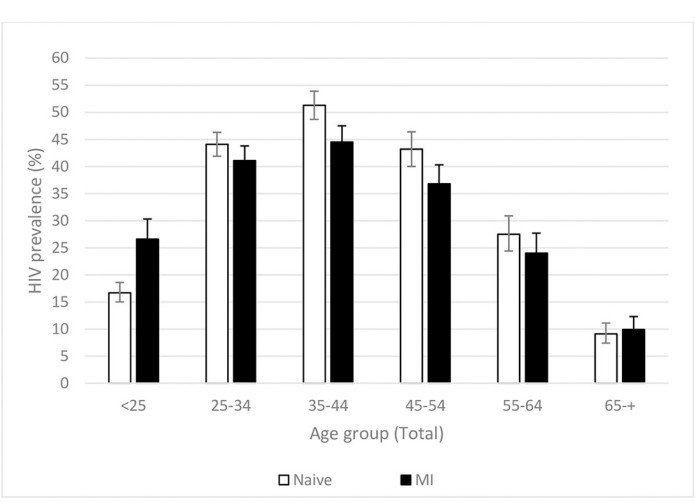

**Fig 3.** Age-specific HIV prevalence according to sex (A, men; B, women; C, total) and population distribution in rural southern Mozambique. HIV prevalence is shown unadjusted and excluding missing values (white bars) and after adjustment for missing values by multiple imputation (MI) and by inverse probability weighting (IPW). Fold-change in prevalence between unadjusted and adjusted values was calculated by dividing the adjusted prevalence estimate by the unadjusted prevalence estimate.

advanced WHO stage and male gender were associated with LTFU at 18 months (SHR 2.47 [95% CI 1.34 to 4.52] and SHR 2.13 [95% CI 1.34 to 3.38], respectively; S2 Fig). In the multivariate model adjusting for age, testing modality, and WHO stage at first visit, only male gender (adjusted SHR 1.89 [95% CI 1.12 to 3.17]) was independently associated with LTFU (S1 Table).

During the 24 months after HIV diagnosis, almost twofold more men died than women (9.33% [46/493] vs. 5.56% [35/629], respectively) with a univariate HR of death for men of 1.72

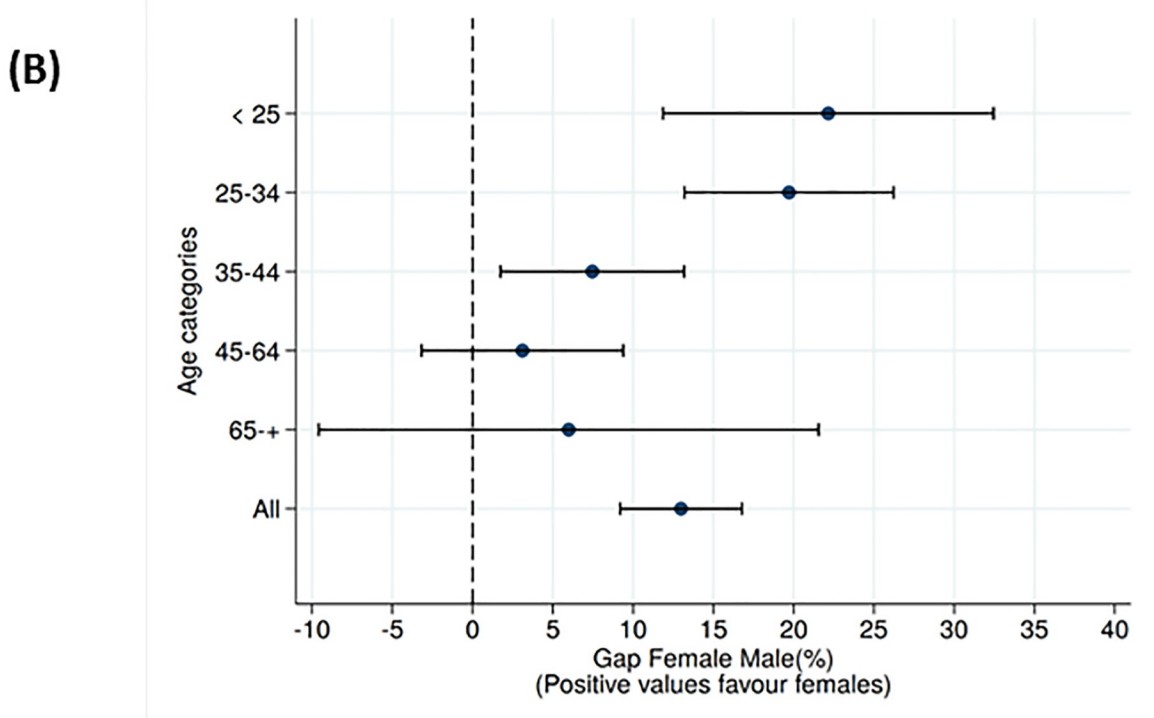

**(A)**

| Age category | Women (95%CI) | Men (95%CI) | Total (95%CI) | Difference (women - men) | p-value |
|---|---|---|---|---|---|
| Total | 88.9 (87.5 to 90.4) | 75.9 (72.4 to 79.5) | 85.4 (84.0 to 86.8) | 13.0 (9.2 to 16.8) | < 0.001 |
| <25 | 87.5 (83.6 to 91.3) | 65.3 (55.6 to 74.9) | 82.8 (79.4 to 86.2) | 22.2 (11.9 to 32.5) | < 0.001 |
| 25-34 | 92.1 (89.9 to 94.3) | 72.4 (66.2 to 78.6) | 86.4 (84.2 to 88.7) | 19.7 (13.2 to 26.2) | < 0.001 |
| 35-44 | 88.4 (85.8 to 91.1) | 81.0 (75.9 to 86.1) | 86.3 (83.9 to 88.7) | 7.5 (1.7 to 13.2) | 0.011 |
| 45-64 | 87.0 (83.8 to 90.1) | 83.9 (78.4 to 89.3) | 86.1 (83.3 to 88.8) | 3.1 (-3.2 to 9.4) | 0.333 |
| >64 | 83.0 (73.8 to 92.3) | 77.0 (64.5 to 89.6) | 81.4 (74.2 to 88.7) | 6.0 (-9.6 to 21.6) | 0.451 |

**Fig 4. Gap in knowledge of HIV positive serostatus between men and women adjusted by multiple imputation method.** (A) Age-specific and sex-specific proportion of HIV-infected individuals who were aware of their serostatus at the time of the survey; (B) the proportional difference in serostatus awareness between the two sexes. The graph shows the point estimates plus 95% confidence intervals (CI) for the percent difference between female and male awareness of their HIV status. The confidence interval for the proportion difference was calculated from the two-sample unpaired z test for each comparison.

(95% CI 1.11 to 2.67) compared to women (S2 Table). When adjusted for age, testing modality, ART initiation, and WHO stage, both WHO stage and older age predicted death. Those with unknown WHO stage had a risk similar to stage I/II (SHR 0.50 95% CI 0.24 to 1.02). Having initiated ART was associated with reduced mortality (adjusted SHR, 0.46 [95% CI 0.23 to 0.91]; S2 Table).

## Discussion

We prospectively assessed progress in reaching the first two targets of the 90-90-90 strategy in HIV-infected men and women in rural southern Mozambique. Estimated proportions of PLWHIV aware of their status (first 90 target) were 75.9% for men and 88.9% for women. Men younger than 35 years significantly lagged behind women of the same age in reaching the first 90 target, and four fold more men than women had WHO stage III/IV AIDS at the first clinical visit. Once enrolled in ART, men had a twofold higher incidence of LTFU at 18 months than women, independent of WHO stage.

Striving toward the first 90 target has revealed many challenges in designing strategies for both efficient and equitable HIV testing [23]. In Manhiça, achievement of the first 90 target in men and women was higher than the UNAIDS estimates for Mozambique of 59% (95% CI 49–70) based on back-calculation of HIV national prevalence. However, there were great age disparities, with the largest for individuals younger than 25 years among whom 87.5% of women and 65.3% of men were aware of their HIV-positive status. Similar findings were reported in community-based studies in Kwazulu Natal, where awareness of HIV-positive status ranged from 53% to 75%, in 2014 [24,25]. Our study supports the the results of the PopArt [26] universal community testing rounds in Zambia and Western Cape of South Africa which showed greater gains in the first 90 target in women than men in 2016 and highlighted difficulties locating men. Some recent interventions, including HIV self testing at the facility or community level have proven successful in narrowing this gap in HIV testing, although linkage to care remains challenging [27,28].

In the Manhiça HBT campaign, although the yield of new HIV diagnoses was higher in men than in women, HTC response rates for both genders were relatively low. The significantly lower response in men (48.7% vs. 62.0% in women) was mainly due to repeated absence from the household and tonon-participation. Men were three times more likely than women to be absent from the household. This could be due partly to the unpredictability of men's presence at home owing to work and mobility compared with women's more predictable schedules from work in the agricultural sector, domestic chores and childcare. Among individuals who were contacted, non-participation were also higher in men than women [29]. Secondary distribution of HIV self tests by women to their male partners might be a promising approach to overcome some of these barriers [30].

High rates of missing data for HTC add uncertainty to estimates of community HIV prevalence. An analysis of 12 HDSS in Southern Africa found that several underestimated HIV prevalence estimates in both genders largely owing to high survey non-response rates [31]. In our survey, we observed a decrease in HIV prevalence after adjustment by MI in men and women of most age groups. However in those aged <25 years, and among women older than 65 years, MI increased the prevalence estimates. Several studies have suggested that higher HIV prevalence estimate in women may be due to an underestimation of prevalence in men due to exclusion of those with missing HIV-serostatus [31]. Our results suggest that younger undiagnosed PLWHIV may be more likely to be absent than older undiagnosed PLWHIV. Thus uncertainty and the direction and magnitude of the missing HIV serostatus bias may differ according to type of non-response and/or to the maturity of the epidemic. Absence due to migration, to

non-participation or due to employment may carry different associations with risk of HIV positivity. Exclusion of those individuals missing HIV status data from estimates of the first 90 target could lead to inaccurate estimations of progress in reaching the 90-90-90 targets. Policy makers should consider a larger range of HIV prevalence estimates, particularly for age and gender-stratified sub-populations.

Throughout the male HIV care cascade, disentangling structural and social factors related to poor outcomes from diagnosis of HIV at advanced stages of infection was difficult. In Manhiça, both men and non-pregnant women linked equally poorly to care, with up to 57% missing CD4 data, similar to observations in neighboring sub-Saharan countries [25,32]. In individuals linked to care, four fold more men than women had WHO stage III/IV AIDS, and 20.6% of men had CD4 <100 cells per μL, in line with recent reports showing that in the Western Cape in South Africa, 39% of men first present with CD4 counts below 200 copies/ mL compared to 25% in women When adjusted for age, WHO stage, and ART status, gender no longer predicted death, whereas age and WHO stage—which are gender-associated variables—were highly associated with increased mortality. PLWHIV with more advanced WHO stage and/or disease are more likely to link to care and initiate ART than healthier PLWHIV [10,33–36]. Indeed, recent findings in Kwazulu Natal show that, once PLWHIV have a CD4 measurement, the probability of ART initiation decreases by 17% for every 100-cell increase in baseline CD4 count [33]. In our population, once PLWHIV reached ART criteria, a high proportion of men and women initiated ART. However, once enrolled, after accounting for deaths and transfers, men were twice as likely to be LTFU within the first 18 months of ART.

This study has several limitations. First, this cohort was recruited in 2014 to 2015, in a pre "Test and Treat" scenario. However, to our knowledge, our results show for the first time the gender gap in the HIV cascade in Mozambique, and provide valuable and hard to obtain population-based data from a high prevalence, low-income country. Secondly, individuals who self-reported a HIV-negative result in the previous three months were not retested, so their HIV status could not be confirmed. However, this group represented only 2.6% of the individuals contacted (284/10,881). In estimating and adjusting age and gender-specific HIV prevalence for missing HIV serostatus we applied MI, assuming that HIV status is missing at random. Studies have suggested this is not the case and thus we cannot exclude selection bias [31,37]. Some authors have suggested Heckman-type selection models to correct for sample selection bias in HIV testing surveys, but many HDSS, including ours are missing data on the selection variables [31]. Manhiça is characterized by a high external migration rate, which reaches up to 230/1000 person-years among men aged 20–25 years [15]. Consequently, we could have overestimated the proportion of LTFU owing to missing data among migrants. Our estimation of the second 90 was based on a cohort of PLHIV diagnosed through three modalities. The fraction of PLHIV diagnosed through each modality may vary according to subnational level highlighting the need for caution in comparing factors associated with the second 90 in different settings.

Our results suggest that estimating indicators of HIV prevalence and of achievement of UNAIDS targets may benefit from subnational community surveys disaggregated sufficiently to detect age-, gender- and geographic-specific service gaps. The work also uncovers the multitude of layers constraining men's access to HIV testing and care, represented as the first 90, where men are physically absent or more likely to decline testing, drop out of care or wait until illness obliges them to seek care. Increased efforts to characterize missing men and their needs will allow us to urgently address the barriers to men accessing care andensure men are not left behind in the UNAIDS 90-90-90 targets achievement.

## Supporting information

**S1 Fig. Progress toward the first 90 target in rural southern Mozambique.** (A) Age-specific and sex-specific proportion of HIV-infected individuals who were aware of their serostatus at the time of the survey; (B) the proportional difference in serostatus awareness between the two sexes. The graph shows the point estimates plus 95% confidence intervals (CI) for the percent difference between female and male awareness of their HIV status.
(DOCX)

**S2 Fig. Cumulative incidence of loss to follow-up (LTFU) after antiretroviral therapy (ART) initiation among men and women in rural southern Mozambique.** Unadjusted cumulative proportion of LTFU after ART initiation over time. The unadjusted sub-distribution hazard ratio (SHR) of LTFU for men vs women was 2.13 (95% CI 1.34 to 3.38).
(DOCX)

**S1 Table. Univariate and multivariable models of factors associated with loss to follow-up at 18 months after initiation of antiretroviral therapy in rural southern Mozambique.**
(DOCX)

**S2 Table. Univariate and multivariable models of factors associated with mortality during 24 months after HIV diagnosis in rural southern Mozambique.**
(DOCX)

## Acknowledgments

The authors gratefully acknowledge the Ministry of Health of Mozambique, our research team, collaborators, and especially all communities and participants involved Also, we want acknowledge Elisabeth Salvo for her contributions.

## Author Contributions

**Conceptualization:** Elisa Lopez-Varela, Orvalho Augusto, Laura Fuente-Soro, Ariel Nhacolo, Denise Naniche.

**Data curation:** Elisa Lopez-Varela, Orvalho Augusto, Laura Fuente-Soro, Charfudin Sacoor, Esmeralda Karajeanes, Paula Vaz.

**Formal analysis:** Elisa Lopez-Varela, Orvalho Augusto, Laura Fuente-Soro, Denise Naniche.

**Funding acquisition:** Denise Naniche.

**Investigation:** Elisa Lopez-Varela, Orvalho Augusto, Denise Naniche.

**Methodology:** Elisa Lopez-Varela, Orvalho Augusto, Laura Fuente-Soro, Charfudin Sacoor, Ariel Nhacolo, Esmeralda Karajeanes, Denise Naniche.

**Project administration:** Laura Fuente-Soro, Isabelle Casavant.

**Resources:** Elisa Lopez-Varela, Esmeralda Karajeanes, Paula Vaz.

**Supervision:** Elisa Lopez-Varela, Laura Fuente-Soro, Charfudin Sacoor, Ariel Nhacolo, Isabelle Casavant, Denise Naniche.

**Validation:** Elisa Lopez-Varela, Orvalho Augusto, Charfudin Sacoor, Ariel Nhacolo, Paula Vaz, Denise Naniche.

**Visualization:** Laura Fuente-Soro.

**Writing – original draft:** Elisa Lopez-Varela.

**Writing – review & editing:** Elisa Lopez-Varela, Orvalho Augusto, Laura Fuente-Soro, Charfudin Sacoor, Ariel Nhacolo, Isabelle Casavant, Esmeralda Karajeanes, Paula Vaz, Denise Naniche.

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
