## [Decision Letter · Decision Letter 0]

25 Aug 2020

PONE-D-20-05705

Quantifying the gender gap in the HIV care cascade in southern Mozambique: Where are the men?

PLOS ONE

Dear Dr. Lopez-Varela,

Thank you for submitting your manuscript to PLOS ONE. After careful consideration, we feel that it has merit but does not fully meet PLOS ONE’s publication criteria as it currently stands. Therefore, we invite you to submit a revised version of the manuscript that addresses the points raised during the review process.

We look forward to receiving your revised manuscript.

Kind regards,

Henry F. Raymond

Academic Editor

PLOS ONE

Journal Requirements:

2.We note that you have indicated that data from this study are available upon request. PLOS only allows data to be available upon request if there are legal or ethical restrictions on sharing data publicly. For information on unacceptable data access restrictions, please see http://journals.plos.org/plosone/s/data-availability#loc-unacceptable-data-access-restrictions.

Additional Editor Comments (if provided):

Dear Author,

The reviewers' comments are many and important. It might be a big and potentially an impossible task to address all of them. However, perhaps you might be able to and high lite a neglected segment of the population affected by HIV, men. Perhaps simplifying and reducing the scope of your inference would strengthen your conclusions.

Reviewers' comments:

Reviewer's Responses to Questions

**Comments to the Author**

1. Is the manuscript technically sound, and do the data support the conclusions?

Reviewer #1: Yes

Reviewer #2: No

2. Has the statistical analysis been performed appropriately and rigorously? 

Reviewer #1: Yes

Reviewer #2: I Don't Know

3. Have the authors made all data underlying the findings in their manuscript fully available?

Reviewer #1: Yes

Reviewer #2: Yes

4. Is the manuscript presented in an intelligible fashion and written in standard English?

Reviewer #1: Yes

Reviewer #2: Yes

5. Review Comments to the Author

Reviewer #1: Thank you for the opportunity to review this well-written, important article which aims to quantify the gender gap in the HIV care cascade in southern Mozambique.

Major comments

In general, as you are interested in gender differences, your results should be presented by gender throughout (particularly Table 1 and Figure 1).

The discussion would benefit from more comment on possible implications of study findings and recommendations, including what has or hasn’t been done to try to engage men in the HIV cascade (see the recent JIAS supplement on Men and HIV).

I think the study has strengths that should be included in the strengths & limitations, especially that this is population-based data from a high prevalence, low-income country.

Minor comments

P2, line 10: disaggregate total by sex.

P2, lines 11-12: ‘Adjusted HIV-community prevalence was estimated using MI’. Move to methods.

Line 19: Once on ART (not in ART)

Line 38: unless you are able to confirm that all deaths among HIV+ are HIV-related, please reword as ‘HIV-positive men have a 41% higher risk of dying than HIV-positive women/

Lines 62 onwards: please add basic information about socio-economic status ie mostly poor, accessing public health system. This may not be clear to a reader unfamiliar with the context.

Line 82: ‘proxy for’ not ‘proxy to’

Line 92: ‘Those with prior enrolment history of HIV care were not visited’

Line 92-3: ‘Since pregnant women followed a specific model of integrated care, they were excluded from the study cohort which assessed the HIV care cascade for non-pregnant adults.’

Lines 112 and elsewhere: consider using an alternative word to ‘refuse’; perhaps you could say they chose not to participate? It sounds less judgmental!

Line 116: add ‘having’ ie ‘and having no existing record in the ePTS’.

Line 125: delete (6 months).

Line 132: unadusted models excluding those with missing data are usually called complete cases.

Line 158 & Table 1: As your primary interest is gender differences, Table 1 should be revised and presented by gender ie responders, male and female columns below; non-responders, male and female columns. This will make gender differences in their characteristics clearer. Also, I would expect a little more detailed reporting from Table 1 (which should include stage and CD4 count), which you’ll be able to give once you revise as suggested. Table 1 should also be presented as close to the text reporting from it as possible.

Line 160: replace ‘were not approached at home’ with ‘were ineligible’.

Line 165: ‘delete ‘as previously described’ – this would fit in a discussion; delete ‘but’.

Line 166: capitalise Absences

Line 168: delete ‘additionally’; Capitalise Of

Table 1: if you report a median age, include median age in the Table.

Headings & footnotes for Tables and Figures should be single spaced.

Lines 191-192: Revise as: ‘Responders differed significantly in gender and age from individuals without HIV status data due to absenteeism, migration and non-participation (Table 1).’

Lines 192-195: Delete from: ‘To account for the … ‘ to ‘refused HIV testing’. Then report the complete case estimates by gender, and report that adjusting for multiple imputation reduced the estimates in all ages except among those <25 years (you can cite the estimates), and among women 65+ years (same).

P12, lines 214 onwards. Figure 1 should be done by sex so that you reporting reflects what we can find in the figure.

Lines 219-222: Delete from ‘To monitor the progress …’ to ‘status. In order to’.

Line 222: Capitalise ‘To account’

Line 244: delete ‘similar proportions to those in Mozambique programmatic reports (44% PICT and 29% VCT)[18].’ This is a comment and should be in the discussion.

Line 274-5: delete ‘gender was borderline associated with death’. I think this is an error, as I cannot see a p=0.064; according to S2 table, the association with male gender did not persist after adjustment (1.32, 0.84-2.09, p=0.230). This should be reported.

Lines 276-278 report incorrect values from Table S2, and these are SHRs not AHRs.

Line 294: revise as: ‘younger than 25 years, among whom …’

Line 297: Revise as :’Our study supports the results of the PopArt …’

Line 298: add “which’ before ‘showed’

Line 301: delete the percentages, these have been cited in the results.

Line 303: consider replacing ‘to refusal’ with ‘choosing not to participate’.

Line 307: replace ‘For’ with ‘Among’

Line 308: add ‘also’ before ‘higher’

Line 313: this is a misinterpretation of Figures 3a-c. There was a decrease, not an increase, in prevalence after MI. Please check and review the paragraph in light of this. I think this should be revised to say: ‘In our survey, we observed a decrease in HIV prevalence after adjustment by MI in men and women of most age groups. However in those aged <25 years, and among women older than 65 years, MI increased the prevalence estimates.

Line 344: replace ‘shown’ with ‘show’. Also, this is not a limit. This is a strength!

Line 364: replace ‘geographical’ with ‘geographic’.

Line 365: reword as ‘The work also uncovers the multitude of layers constraining men’s access to HIV testing and care, represented as the first 90.’

Line 367: I think you need a different closing sentence. The need is not just to characterise missing men and their needs, but to urgently address barriers to men accessing care.

Reviewer #2: The authors submitted the manuscript entitled Quantifying the gender gap in the HIV care cascade in southern Mozambique: Where are the men? UNAIDS has set an ambitious global strategy for reaching 2020 goals towards ending the HIV epidemic. These include achieving 90% of people living with HIV (PLWHIV) aware of their HIV status, 90% of those will be on antiretroviral therapy (ART), and of those, 90% will reach viral suppression. These are described as the 90-90-90 targets to end the HIV epidemic. The authors note that HIV-infected men have higher rates of delayed diagnosis, reduced antiretroviral treatment (ART) retention and mortality than women. They wanted to assess, by gender, the first two UNAIDS 90 targets in rural southern Mozambique.

The authors note that there are limited statistically sampled population-based data to estimate the achievement of the UNAIDS targets. Therefore, they proposed to analyze and calculate these estimates from a large prospective cohort recruited from Manhiça district, a rural area in Mozambique with a high HIV burden of disease. The cohort included adults enrolling with new HIV diagnosis between May 2014-June 2015 from clinic and home-based testing (HBT) identification and recruitment methods. They hypothesized that there would be significant gender differences between steps of the HIV-cascade in this rural region of Mozambique.

The authors employed multiple separate non-randomly selected samples to perform analysis. For example, they assessed respondents among 11,773 adults randomized in HBT. However, the response rate before HIV testing was 48.7% among eligible men and 62.0% among women. Therefore, this sample was self- selected and biased. They also recruited participants using two other methods; voluntary counselling and testing (VCT), provider-initiated counselling and testing (PICT) recruited at two clinic-based venues. The authors tried to accommodate those concerns by adjusting HIV-prevalence using multiple imputation (MI), but there were significant numbers of missing data, which may make imputation simulation models unstable. Likewise, statistical analyses based on non-randomly selected samples can lead to erroneous conclusions. Statistical methods such as the Heckman correction can provide a means of correcting for non-randomly selected samples, but these method were not employed.

Despite these limitations, the authors estimated proportion of HIV-infected individuals aware of their status was 75.9% for men and 88.9% for women. In individuals <25 years, they estimated a 22.2% difference in awareness of serostatus rates between men and women. Among individuals eligible for ART, similar proportions of men and women initiated treatment (81.2% and 85.9%, respectively). Fourfold more men than women were in WHO stage III/IV AIDS at first clinical visit. Once in ART, men had a twofold higher 18-20 month loss to follow-up rate than women.

The authors appropriately conclude that estimating indicators of HIV prevalence and of national or regional outcomes towards the UNAIDS90-90-90 targets may benefit from regional or national population based community surveys to detect age-, gender- and geographical-specific service gaps. Likewise, increased efforts to characterize under-reported and missing data especially involving men, would help identify men’s’ needs and ensure men are not left behind in the UNAIDS 90-90-90 target goals.

6. PLOS authors have the option to publish the peer review history of their article (what does this mean?). If published, this will include your full peer review and any attached files.

Reviewer #1: No

Reviewer #2: No

---

## [Author Response · Author response to Decision Letter 0]

22 Oct 2020

Specific responses to reviewers and editor

Editor

The reviewers' comments are many and important. It might be a big and potentially an impossible task to address all of them. However, perhaps you might be able to and highlite a neglected segment of the population affected by HIV, men. Perhaps simplifying and reducing the scope of your inference would strengthen your conclusions

• We thank the editor for this comment. We have followed the suggestion and highlighted men as a neglected segment of population and simplified the inferences throughout the discussion. Webelieve the manuscript has improved and the conclusions have been strengthened. 

Reviewer #1:

Major comments

1. In general, as you are interested in gender differences, your results should be presented by gender throughout (particularly Table 1 and Figure 1).

• We have followed the recommendation. For Figure 1, we have added the number of female and male participants at each step of the cascade in brackets. We have added this information for the critical steps while avoiding excessive information which would make the figure difficult to read. The objective of the next Figure (Fig 2) is in fact to show part of that same information but specifically looking at gender differences and presenting the 95%CI. 

• Table 1 has been modified according to the recommendation

2. The discussion would benefit from more comment on possible implications of study findings and recommendations, including what has or hasn’t been done to try to engage men in the HIV cascade (see the recent JIAS supplement on Men and HIV).

• Thank you for this comment. We acknowledge that the original manuscript was submitted eight months ago. Since then, there have been a number of publications available on the topic. We have updated the references and discussion accordingly, including a slight change in the title in line with current narrative which attempts to view men as part of the solution rather than “men as a problem” . 

3. I think the study has strengths that should be included in the strengths & limitations, especially that this is population-based data from a high prevalence, low-income country

• This has been added in the conclusion section and the abstract- see line 30 and 421

Minor comments

P2, line 10: disaggregate total by sex.

• This has been added- see line 14.

P2, lines 11-12: ‘Adjusted HIV-community prevalence was estimated using MI’. Move to methods.

• We have made the correction. See line 12.

Line 19: Once on ART (not in ART)

• This has been corrected. 

Line 38: unless you are able to confirm that all deaths among HIV+ are HIV-related, please reword as ‘HIV-positive men have a 41% higher risk of dying than HIV-positive women/

• This has been corrected. 

Lines 62 onwards: please add basic information about socio-economic status ie mostly poor, accessing public health system. This may not be clear to a reader unfamiliar with the context.

• We have added a sentence and a reference : “The majority of the population are rural and most are en-gaged in subsistence farming, There is a predominance of young people and nearly half of all women and a quarter of all men are illiterate.[16]”- Line 79-81

Line 82: ‘proxy for’ not ‘proxy to’

• This has been corrected. 

Line 92: ‘Those with prior enrolment history of HIV care were not visited’

• This has been corrected. 

Line 92-3: ‘Since pregnant women followed a specific model of integrated care, they were excluded from the study cohort which assessed the HIV care cascade for non-pregnant adults.’

• This has been corrected. 

Lines 112 and elsewhere: consider using an alternative word to ‘refuse’; perhaps you could say they chose not to participate? It sounds less judgmental!

• We have corrected this throughout the paper.

Line 116: add ‘having’ ie ‘and having no existing record in the ePTS’.

• This has been corrected. 

Line 125: delete (6 months).

• We have deleted these words.. 

Line 132: unadusted models excluding those with missing data are usually called complete cases.

• We have added the term-line 153

Line 158 & Table 1: As your primary interest is gender differences, Table 1 should be revised and presented by gender ie responders, male and female columns below; non-responders, male and female columns. This will make gender differences in their characteristics clearer. 

And 

Table 1: if you report a median age, include median age in the Table.

• We have modified Table1 following this suggestion. – line 207.

• We would like to note that he objective of Table 1 is indeed to present how gender influenced the proportion of responders/non responders. Thus, the denominator for the table (N=9093), excludes ineligible/pregnant women and is thus different from the overall study population of 11773 mentioned in line 158. As such, the median age mentioned in line 159 refers to this overall population of 11773 and not to the responder/non responder groups. We have clarified this in the text (line 180). For Table 1, we believe it is more appropriate to show age categories rather than median, as these same age groups are then showed in the next figures when MI is used. Reporting only the median fails to show how the distribution of responders/non responders vary across more extreme age groups. 

Also, I would expect a little more detailed reporting from Table 1 (which should include stage and CD4 count), which you’ll be able to give once you revise as suggested. Table 1 should also be presented as close to the text reporting from it as possible.

• We agree with the reviewer but Table 1 presents differences in gender for responders/non responders. Non responders were, by definition, those not found or not tested, so there is no baseline HIV clinical information available for them. 

Line 160: replace ‘were not approached at home’ with ‘were ineligible’.

• We have modified accordingly.

Line 165: ‘delete ‘as previously described’ – this would fit in a discussion; delete ‘but’.

• We have modified accordingly

Line 166: capitalise Absences

• This has been corrected.

Line 168: delete ‘additionally’; Capitalise Of

• This has been corrected.

Headings & footnotes for Tables and Figures should be single spaced.

• We have modified headings and footnotes.

Lines 191-192: Revise as: ‘Responders differed significantly in gender and age from individuals without HIV status data due to absenteeism, migration and non-participation (Table 1).’

• We have modified accordingly.

Lines 192-195: Delete from: ‘To account for the … ‘ to ‘refused HIV testing’. Then report the complete case estimates by gender, and report that adjusting for multiple imputation reduced the estimates in all ages except among those <25 years (you can cite the estimates), and among women 65+ years (same).

• We have modified accordingly.

P12, lines 214 onwards. Figure 1 should be done by sex so that you reporting reflects what we can find in the figure.

• Please see comment above. We have also added a reference to Fig 2 in line 252, which was missing. Fig 2 shows the differences in prevalence, with the corresponding 95%CI. 

Lines 219-222: Delete from ‘To monitor the progress …’ to ‘status. In order to’.

Line 222: Capitalise ‘To account’

• We have modified accordingly.

Line 244: delete ‘similar proportions to those in Mozambique programmatic reports (44% PICT and 29% VCT)[18].’ This is a comment and should be in the discussion.

• This has been corrected.

Line 274-5: delete ‘gender was borderline associated with death’. I think this is an error, as I cannot see a p=0.064; according to S2 table, the association with male gender did not persist after adjustment (1.32, 0.84-2.09, p=0.230). This should be reported.

• We thank the reviewers for pointing this out. We have corrected the text accordingly. Line 310

Lines 276-278 report incorrect values from Table S2, and these are SHRs not AHRs.

• We thank the reviewer for pointing this out. We have corrected this section. 

Line 294: revise as: ‘younger than 25 years, among whom …’

• This has been corrected.

Line 297: Revise as :’Our study supports the results of the PopArt …’

• We have modified.

Line 298: add “which’ before ‘showed’

• This has been corrected.

Line 301: delete the percentages, these have been cited in the results.

• We have followed the suggestion.

Line 303: consider replacing ‘to refusal’ with ‘choosing not to participate’.

• We have changed accordingly. 

Line 307: replace ‘For’ with ‘Among’

• We have changed accordingly. 

Line 308: add ‘also’ before ‘higher’

• We have changed accordingly. 

Line 313: this is a misinterpretation of Figures 3a-c. There was a decrease, not an increase, in prevalence after MI. Please check and review the paragraph in light of this. I think this should be revised to say: ‘In our survey, we observed a decrease in HIV prevalence after adjustment by MI in men and women of most age groups. However in those aged <25 years, and among women older than 65 years, MI increased the prevalence estimates.

• We have modified the text following the suggestion. See line 357.

Line 344: replace ‘shown’ with ‘show’. Also, this is not a limit. This is a strength!

• We have changed the text- 

Line 364: replace ‘geographical’ with ‘geographic’.

• This has been corrected. 

Line 365: reword as ‘The work also uncovers the multitude of layers constraining men’s access to HIV testing and care, represented as the first 90.’

• We have reworded the sentence. 

Line 367: I think you need a different closing sentence. The need is not just to characterise missing men and their needs, but to urgently address barriers to men accessing care.

• Thank you, We have rephrased the closing sentence following the suggestion. 

Reviewer #2: 

The authors submitted the manuscript entitled Quantifying the gender gap in the HIV care cascade in southern Mozambique: Where are the men? UNAIDS has set an ambitious global strategy for reaching 2020 goals towards ending the HIV epidemic. These include achieving 90% of people living with HIV (PLWHIV) aware of their HIV status, 90% of those will be on antiretroviral therapy (ART), and of those, 90% will reach viral suppression. These are described as the 90-90-90 targets to end the HIV epidemic. The authors note that HIV-infected men have higher rates of delayed diagnosis, reduced antiretroviral treatment (ART) retention and mortality than women. They wanted to assess, by gender, the first two UNAIDS 90 targets in rural southern Mozambique.

The authors note that there are limited statistically sampled population-based data to estimate the achievement of the UNAIDS targets. Therefore, they proposed to analyze and calculate these estimates from a large prospective cohort recruited from Manhiça district, a rural area in Mozambique with a high HIV burden of disease. The cohort included adults enrolling with new HIV diagnosis between May 2014-June 2015 from clinic and home-based testing (HBT) identification and recruitment methods. They hypothesized that there would be significant gender differences between steps of the HIV-cascade in this rural region of Mozambique.

The authors employed multiple separate non-randomly selected samples to perform analysis. For example, they assessed respondents among 11,773 adults randomized in HBT. However, the response rate before HIV testing was 48.7% among eligible men and 62.0% among women. Therefore, this sample was self- selected and biased. They also recruited participants using two other methods; voluntary counselling and testing (VCT), provider-initiated counselling and testing (PICT) recruited at two clinic-based venues. The authors tried to accommodate those concerns by adjusting HIV-prevalence using multiple imputation (MI), but there were significant numbers of missing data, which may make imputation simulation models unstable. Likewise, statistical analyses based on non-randomly selected samples can lead to erroneous conclusions. Statistical methods such as the Heckman correction can provide a means of correcting for non-randomly selected samples, but these method were not employed. 

Despite these limitations, the authors estimated proportion of HIV-infected individuals aware of their status was 75.9% for men and 88.9% for women. In individuals <25 years, they estimated a 22.2% difference in awareness of serostatus rates between men and women. Among individuals eligible for ART, similar proportions of men and women initiated treatment (81.2% and 85.9%, respectively). Fourfold more men than women were in WHO stage III/IV AIDS at first clinical visit. Once in ART, men had a twofold higher 18-20 month loss to follow-up rate than women.

The authors appropriately conclude that estimating indicators of HIV prevalence and of national or regional outcomes towards the UNAIDS90-90-90 targets may benefit from regional or national population based community surveys to detect age-, gender- and geographical-specific service gaps. Likewise, increased efforts to characterize under-reported and missing data especially involving men, would help identify men’s’ needs and ensure men are not left behind in the UNAIDS 90-90-90 target goals.

• We thank the reviewer for his comments and we fully agree with the reviewer regarding the limitations of the models used. 

• For this reason, we added a section on the discussion- limitation section- explaining potential bias associated with model used and reason why we were not able to use Heckman-type selection models with data available. See line 401-406: “In estimating and adjusting age and gender-specific HIV prevalence for missing HIV serostatus we applied MI, assuming that HIV status is missing at random. Studies have suggested this is not the case and thus we cannot exclude selection bias[31,37]. Some authors have suggested Heckman-type selection models to correct for sample selection bias in HIV testing surveys, but many HDSS, including ours are missing data on the selection variables[31]”

---

## [Editor Report · Decision Letter 1]

4 Jan 2021

Quantifying the gender gap in the HIV care cascade in southern Mozambique: we are missing the men

PONE-D-20-05705R1

Dear Dr. Lopez-Varela,

We’re pleased to inform you that your manuscript has been judged scientifically suitable for publication and will be formally accepted for publication once it meets all outstanding technical requirements.

Kind regards,

John S Lambert

Academic Editor

PLOS ONE

Additional Editor Comments (optional):

I have reviewed the revised manuscript which has incorporated the comments and suggestions of reviewer 1 and reviewer 2, and would advice accepting, as all changes have been made and the publication is relevant to ongoing issues with HIV treatment uptake in Africa
---

## [Editor Report · Acceptance letter]

8 Jan 2021

PONE-D-20-05705R1 

Quantifying the gender gap in the HIV care cascade in southern Mozambique: we are missing the men 

Dear Dr. Lopez-Varela:

I'm pleased to inform you that your manuscript has been deemed suitable for publication in PLOS ONE. Congratulations! Your manuscript is now with our production department. 

Kind regards, 

on behalf of

Dr. John S Lambert 

Academic Editor

PLOS ONE